# An Update on the Role of Anti-EGFR in the Treatment of Older Patients with Metastatic Colorectal Cancer

**DOI:** 10.3390/jcm11237108

**Published:** 2022-11-30

**Authors:** Gerardo Rosati, Michele Montrone, Carmen Pacilio, Alfredo Colombo, Giuseppe Cicero, Fernando Paragliola, Angelo Vaia, Luigi Annunziata, Domenico Bilancia

**Affiliations:** 1Medical Oncology Unit, “San Carlo” Hospital, 85100 Potenza, Italy; 2Medical Thoracic Oncology Unit, IRCCS Istituto Tumori “Giovanni Paolo II”, 70124 Bari, Italy; 3Medical Breast Cancer Department, IRCCS Istituto Tumori “G. Pascale”, 80131 Napoli, Italy; 4Medical Oncology Unit, CDC Macchiarella, 90138 Palermo, Italy; 5Medical Oncology, Università degli Studi di Palermo, 90133 Palermo, Italy

**Keywords:** cetuximab, metastatic colorectal cancer, older patients, panitumumab

## Abstract

Although colorectal cancer is increasingly being diagnosed in older patients, their number is largely underrepresented in phase II or III clinical trials. Consequently, guidelines and the SIOG recommendations are not sufficiently clear regarding the treatment of these patients, particularly when chemotherapy is combined with monoclonal antibodies (bevacizumab, cetuximab, and panitumumab). Targeted therapy based on the use of anti-epidermal growth factor receptors (EGFRs) is conditioned by the potential for increased toxicity, making it more difficult to treat an older, rat sarcoma virus (RAS) and B rapidly accelerated fibrosarcoma (BRAF) wild-type patient. In light of a more detailed characterization of the older population, modernly differentiable between fit, vulnerable, or frail patients on the basis of the comprehensive geriatric assessment, and of the analysis of more recent studies, this review fully collects data from the literature, differentiating the results on functional status patients.

## 1. Introduction

It is estimated that nearly 2.0 million new cases of colorectal cancer (CRC) were diagnosed in 2020 and 935,000 patients will die from it, causing approximately one in 10 deaths from cancer [1]. For all intents and purposes, this disease could be considered to be a marker of socioeconomic development, which could explain how higher incidence and mortality rates are found in Western countries [2], and also why this tumor has increasingly become a characteristic of older patients following the lengthening of the average lifespan, longer than in many regions of Africa and South America [3]. For example, in Italy, in 2020, approximately 28,000 new cases of CRC were recorded in patients aged ≥70 years [4].

Nevertheless, the treatment of older patients is not sufficiently standardized as they are not adequately represented in randomized clinical trials, due to inclusion criteria (comorbidity, hepatorenal insufficiency, and performance status), foreclosures by oncologists who fear serious side effects, and the lack of autonomy and adequate caregivers support [5].

Although the International Society of Geriatric Oncology (SIOG) has defined an individual aged >70 as an older person [6], this criterion now appears more than arbitrary. Thanks to the progress of socioeconomic and cultural conditions and thus, to improvements in eating habits, it is sometimes difficult to find a correspondence between biological and physiological age [7]. Thus, on the one hand, we can classify patients without burdensome comorbidities, optimal functional abilities, and favorable psychosocial conditions as “fit” patients and treatable similar to their younger counterparts. On the other hand, patients with poor performace status (PS), loss of autonomy, and various comorbidities are considered to be “frail” patients and they should be excluded from any type of therapy. The majority of older patients, however, fall into a middle range, i.e., that of “vulnerable” subjects, for whom, from time to time, less intensive and adapted therapies could be assigned on a case-by-case basis [8].

The categorization of older patients that has been recommended by the SIOG is based on the use of a comprehensive geriatric assessment (CGA) that tracks different domains (physical and mental health, functional, social, and environmental issues) [9]. However, this instrument can be difficult to apply as it is both time-consuming and resource intensive, and therefore, rapid screening tools such as the Geriatric-8 (G8) have been proposed [10], at least to immediately identify fit subjects and propose only those with G8 scores < 14/17 for CGA. The latter are at potential risk of worsening their functional status and it would be necessary to have the collaboration of a dedicated geriatrician for their treatment.

This premise is necessary to understand why older patients with metastatic CRC (mCRC) may have less substantial benefits from chemotherapy than younger patients. Many older patients might not benefit from drugs such as oxaliplatin, irinotecan, and fluoropyrimidines which, especially if used sequentially and in full doses, allow a gain in survival [11]. While it has long been pointed out that adding monoclonal antibodies (bevacizumab, cetuximab, and panitumumab) to chemotherapy resulted in an improvement in response rate (RR) and overall survival (OS) [12,13,14], this benefit is less evident in the setting of older patients, especially if they are vulnerable. In recent years, randomized clinical trials and retrospective analyses have clarified the potential benefit of bevacizumab in these patients [15,16]. However, this has not happened in the same manner with the use of epidermal growth factor receptor (EGFR) inhibitors, mainly due to the scarcity of dedicated clinical trials.

This review takes stock of the data from studies with the combined use of chemotherapy or not and anti-EGFR by dividing the results among the so-called fit, frail, and vulnerable older patients.

## 2. Methods

We carried out a fine literature research to find relevant publications from 2007 utilizing the PubMed database (www.ncbi.nlm.nih.gov/pubmed/, accessed on 30 September 2022) by entering the following terms: “metastatic colorectal cancer”, “cetuximab”, “panitumumab”, and “older patients”. Full-text manuscripts reporting data on these items were identified and reviewed in detail. In addition, we performed a general excursus of the principal oncology congresses to identify abstracts reporting these data and published between 1 January 2020 and September 2022.

## 3. Anti-EGFR Inhibitors and Fit Patients

As reported above, the SIOG recommendations do not take into account biological age as a determining factor for the treatment of fit older patients [6]. Accordingly, these patients can receive doublet combination therapy with cetuximab or panitumumab (Table 1). However, although selected for inclusion in clinical trials, they appear to derive similar benefits to those for younger patients in terms of RR and progression-free survival (PFS) from the use of these treatments, but there is still no evidence to clearly demonstrate that this results in significant gains for each fit patient for better survival and acceptable quality of life (QoL).

### 3.1. OPUS and CRYSTAL Studies

A pooled analysis of 845 randomized patients in the CRYSTAL and OPUS studies reported that the addition of cetuximab to FOLFIRI or FOLFOX as first-line therapy of patients with wild-type (wt) significantly improved overall response (*p* < 0.0001), PFS (*p* < 0.0001), and OS (*p* < 0.0062) as compared with chemotherapy alone [17]. Folprecht et al. subsequently updated this analysis for OS, PFS, and safety by comparing older patients (age ≥70 years) with younger ones (<70 years) [26]. Although the number of older patients was limited to 78 subjects treated with chemotherapy plus cetuximab and 67 patients treated with chemotherapy alone, PFS and OS were shown to increase with the addition of cetuximab in both older and younger patients and these differences were not statistically significant in the first group (*p* = 0.78 and *p* = 0.38 for PFS and OS, respectively). There was no difference in 60-day mortality rates between the two groups treated with and without cetuximab. However, toxicity appeared to increase in patients >70 years, and in particular grade 3/4 adverse events (AEs) increased with the addition of cetuximab (neutropenia 33.3% and diarrhea 23.1%). Rates of severe diarrhea could be of particular concern given the potential risks of dehydration and also because they are more difficult to manage in older adults and require more hospital admissions than younger patients.

### 3.2. PRIME Study

The randomized phase III PRIME study demonstrated that panitumumab plus FOLFOX significantly increased PFS (10.0 vs. 8.6 months, *p* = 0.01) compared with chemotherapy alone in the first-line treatment of wt Kirsten rat sarcoma virus (KRAS) patients [14]. At the final data analysis and with a median follow-up time of 80.0 weeks, also secondary endpoints such as RR (57% vs. 48%, *p* = 0.02) and OS (23.8 vs. 19.4 months, *p* = 0.03) confirmed the superiority of the addition of panitumumab to chemotherapy. A subsequent exploratory analysis was conducted when ≥80% of patients had an OS event and both efficacy and safety were evaluated for patients with rat sarcoma virus (RAS) wt by baseline age (<65, ≥65 and ≤75, >75 years [27]. Four hundred and ninety-nine patients were included in this ancillary age analysis. For PFS, the hazard ratio (HR) was 0.66 (95% CI 0.52–0.83) in patients < 65 years (*n* = 316) and 0.88 (95% CI 0.65–1.19) in patients ≥ 65 years (*n* = 89). For OS, the HR was 0.75 (95% CI 0.58–0.96) in patients < 65 years and 0.80 (95% CI 0.58–1.09) in patients ≥ 65 years. Patients 75 years of age and older did not have a greater benefit from combination chemotherapy with panitumumab. Although this subgroup consisted of only 34 subjects, and taking into account that often the older patients have a PS of 2, the combination of panitumumab with FOLFOX should be used with caution in very old patients according to the results of this trial. In addition, PRIME data showed an increase in grade 3/4 toxicities associated with panitumumab: skin toxicity (37%), diarrhea (18%) and hypomagnesemia (7%). Although patient-reported QoL was not altered by the addition of panitumumab to FOLFOX, it is undeniable that these toxicities may be more difficult to manage in the older patients.

### 3.3. FIRE-3 Study

This open-label, randomised, phase III trial, recruited 592 patients with KRAS exon 2 wt tumors and aged 18–75 years to receive FOLFIRI plus cetuximab or FOLFIRI plus bevacizumab. The primary endpoint was objective RR analyzed by intention to treat [18]. Although there were no differences in RR between the two arms (62% in the cetuximab group and 58% in the bevacizumab group), a post-hoc analysis of tumour dynamics in the final RAS wt subgroup of this trial, which included 400 patients, showed a better median OS in the FOLFIRI plus cetuximab group than in the FOLFIRI plus bevacizumab group (33.1 months (95% CI 24.5–39.4) vs. 25.0 months (23.0–28.1), HR 0.70 (0.54–0.90), *p* = 0.0059) [28]. Unfortunately, we do not have any data in this regard concerning the age-related differences of patients, as opposed to the original study where it is reported that approximately 45% of them in each arm were over 65 years of age. A nonsignificant benefit in terms of OS (secondary objective) was referred here, regardless of age class, in favor of the combination of cetuximab plus chemotherapy compared to bevacizumab plus chemotherapy: HR 0.75 (95% CI 0.56–1.01) for the 318 patients ≤65 years, and HR 0.80 (95% CI 0.58–1.09) for the 274 patients >65 years.

### 3.4. Observer Study

The Observer is a multicentre, prospective study aimed at investigating the QoL, safety and efficacy of first-line chemotherapy in combination with cetuximab in patients with wt KRAS mCRC [19]. The results of this observational trial demonstrate that the therapy does not have a negative impact on QoL in the routine clinical setting when patients receive close monitoring plus prophylactic or reactive management of skin reactions. A subsequent analysis of this study, comprising 84 fit over seventy-year-olds among the initial 225 patients, with no difference in any of the clinical and pathological features compared to younger ones, showed no difference in PFS and disease control rates (DCR) between the two groups [29]. Median OS was higher in patients < 70 (27 months, 95% CI: 22.7–31.27) than in patients ≥ 70 (19 months, 95% CI: 14.65–23.35) (*p* = 0.002), which was likely due to higher proportions of metastatic resection (27.0% vs. 8.3%, *p* = 0.001) and utilization of second-line therapy in the younger group (58.9% vs. 42.9%, *p* = 0.028). In addition, cetuximab therapy caused a similar incidence of side effects and impact on QoL in older and younger patients, which suggests that fit older patients with mCRC can be safely treated with a cetuximab-based therapy, as QoL and safety profile do not seem to be affected by age.

### 3.5. The ARCAD Database

The Aide de Recherche en Cancerologie Digestive (ARCAD) database included patients receiving first-line doublet chemotherapy plus cetuximab (*n* = 1,191) or doublet chemotherapy alone (*n* = 729) from seven randomized trials with the aim of investigating the prognostic and predictive effect of age (≥70 vs. <70) [20]. The results of this analysis appear to be difficult to interpret. Older (vs younger) patients receiving chemotherapy plus cetuximab had similar PFS (8.7 vs. 10.3 months, HR = 1.20, *p* = 0.107) but inferior OS (21.3 vs. 26.3, HR = 1.36, *p* = 0.011). When PFS and OS were compared by age groups between patients who had received only chemotherapy and those who received its combination with cetuximab, the results confirmed the benefit in PFS (11.2 vs. 8.9, HR = 0.70, *p* < 0.001) and in OS (23.9 vs. 20.3, HR = 0.82, *p* = 0.008) exclusively in the younger patiens, but not in the older ones. Although age groups (<70 vs. ≥70) appeared well balanced and did not differ in sex, number of metastasis, and liver or peritoneum metastasis, these differences in efficacy were explained by the authors that patients ≥ 70 years old were more likely to have prognostically more unfavorable features such as ECOG PS ≥ 1, tumor in the right colon, and metastasis in lungs than those < 70. Older patients (vs younger) had no difference in grade 3 or 4 AEs for neutropenia/leukopenia, diarrhea or nausea/vomiting.

### 3.6. Bouchahda et al. Study

This French retrospective study evaluated the efficacy and safety of cetuximab plus irinotecan in 56 patients aged ≥ 70 years [21]. Their median age was 76 years (range: 70–84 years) and the PS was 0–1 in 83% of cases. Nonetheless, the patients had received a mean number of three previous lines of chemotherapy and 48 of them (86%) had been exposed to fluoropyrimidines, irinotecan and oxaliplatin. While cetuximab was administered weekly on average eight times per patient, irinotecan 180 mg/m^2^ was infused every 2 weeks and in combination in 96% of cases. The efficacy data appeared more than encouraging and the authors concluded that older patients could also benefit from this combination, albeit heavily pretreated with chemotherapy. The toxicities appeared to be comparable to those already documented in other similar trials. Limit of this study is that it recruited patients between June 2004 and August 2005, consequently the RAS status was not determined.

### 3.7. Peeters et al. Study

This large randomized phase III trial recruited nearly 600 patients with mCRC, wt for KRAS, PS between 0 and 2, and previously treated with a line of therapy to receive FOLFIRI plus panitumumab or chemotherapy alone [22]. The primary endpoints of the study were PFS and OS. When panitumumab was added to chemotherapy, a significant improvement in PFS was observed (HR = 0.73, *p* = 0.004); median PFS was 5.9 months for panitumumab-FOLFIRI vs. 3.9 months for FOLFIRI. A nonsignificant trend toward increased OS was observed; median OS was 14.5 months vs. 12.5 months, respectively (HR = 0.85, *p* = 0.12); RR improved to 35% vs. 10% with the addition of panitumumab. AE rates were generally comparable across arms with the exception of known toxicities associated with anti-EGFR therapy. A prespecified subgroup analyses from this study analyzed PFS and OS by randomisation stratification factors (ECOG PS), prior bevacizumab or oxaliplatin, and covariates at baseline such as age [30]. PFS improvements were seen in the panitumumab plus FOLFIRI group in all other subgroups assessed except for those with ECOG PS 2 (*n* = 21), to which older patients could more easily belong. OS analyses gave similar results, but significant OS benefits were seen in panitumumab plus FOLFIRI vs. FOLFIRI-treated patients who had received prior oxaliplatin or prior bevacizumab. Grade 3/4 AEs were more frequent with panitumumab than with chemotherapy alone, but there were no substantial differences based on age: 77% vs. 50% for patients <65 years and 70% vs. 54% for patients. patients ≥65 years.

### 3.8. Spanish Group for Digestive Tumor Therapy (TTD) Studies

Two pioneering studies by this Spanish group investigated the role of cetuximab in older patients [31,32]. The first of these studies evaluated the efficacy and safety of cetuximab given as a single agent in 41 patients and in first-line of therapy. Patients had a mean age of 76 years and a Karnofsky PS of 80–100. Patient compliance was optimal as the median dose intensity was 245.5 mg/m^2^/week and drug administration was postponed due to skin toxicity in only 26.8% of cases. However, the authors explained that this could be attributed to the mean duration of treatment which was only 57 days. Although the results in terms of RR, time to progression (TTP) and OS may appear disappointing, patients were not screened for RAS mutation status because the study was initiated in 2005, when it was not yet known that only wt patients could access to anti-EGFR treatment. A post hoc analysis, performed on 23 patients for whom tumor tissue was available to determine KRAS status, showed that seven of 18 wt KRAS patients were progression-free at 12 weeks, while four of five KRAS mutant patients progressed during the same period. Skin toxicity was in line with that reported by other studies. Although the authors found a correlation between efficacy and skin rash, the small size, low number of responsive patients, and lack of statistical significance (*p* = 0.163) did not allow for definitive conclusions on this statement. This study has the merit of having shown that cetuximab has intrinsic activity and could be used as a first-line treatment in older patients who refuse chemotherapy and require therapy with limited toxicity.

In the second study, cetuximab was initially administered with capecitabine at a dose of 1250 mg/m^2^ twice daily (bid), 2 weeks on and 1 week off. KRAS status was verified only in a proportion (88%) of the 66 enrolled patients. The protocol was amended after the inclusion of the first 27 patients by reducing the dose of capecitabine to 1000 mg/m^2^ bid following severe paronychia, found in 29.6% of patients. Fourteen of the 29 patients with wt KRAS tumors responded (48.3%), compared to six of the 29 patients with mutant KRAS tumors (20.7%). The median PFS for wt KRAS patients was significantly longer than that for mutant KRAS tumors (8.4 months vs. 6.0 months, *p* = 0.024). The authors justified the high grade of skin toxicity considering the possible negative interaction between cetuximab and capecitabine, which emerged already in the COIN study [33]. These data would make the combination of anti-EGFRs with capecitabine inadvisable. Other toxicities, such as eye disorders, nausea, astenia, dry skin, anorexia and hypomagnesaemia, were grade 1/2, while only one case of grade 4 neutropenia was documented.

### 3.9. EREBUS

This French study enrolled 389 wt KRAS patients initiating a first-line therapy with cetuximab in combination with irinotecan or oxaliplatin for 56.8% and 38.5% of cases, respectively [23]. The objectives of this observational study were to evaluate the metastasis resection rate, safety and OS in patients treated in real clinical practice. ECOG PS was ≤1 and the liver was the most frequent metastatic site (*n* = 146 exclusively, *n* = 149 not exclusively, *n* = 94 non liver only). The median duration of use of cetuximab was 4.8 months. The results underlined the important advantage for those undergoing liver resection since the median PFS was 9.2 months for the total cohort and 13.0 for those resected, while the median OS was 23.0 months for the total cohort, but was not reached after 36 months for those who were resected. A total of 116 patients (29.8%) aged > 70 years were subsequently analyzed [24]. Older patients were relatively comparable to younger patients for sex and PS (62.1% men vs. 69.6% women (*p* = 0.15); ECOG 0–1: 72.4% vs. 80.2% (*p* = 0.12)). The median duration of cetuximab treatment was significantly shorter for older patients (3.7 vs. 5.3 months, *p* = 0.03). The frequency of association with irinotecan-based (54.3% vs. 56.8%, *p* = 0.64) or oxaliplatin-based (36.2% vs. 38.5%, *p* = 0.64) chemotherapy and the incidence of any grade adverse events were not statistically different (haematological (91.4% vs. 90.8%, *p* = 0.87), gastrointestinal (79.3% vs. 85.7%, *p* = 0.12), dermatological (77.6% vs. 85.0%, *p* = 0.08) and neurological (37.9% vs. 47.6%, *p* = 0.08)). No differences emerged in PFS (9.5 vs. 9.2 months) and RR (46.5% vs. 56.7%) between older and younger patients. Results from the EREBUS cohort confirm that fit older patients can tolerate doublet chemotherapy plus anti-EGFR in the same way as younger patients and reap the same benefits.

### 3.10. Abdelwahab et al. Study

This Egyptian phase II study enrolled 49 older patients (≥65 years old) with a median age of 69 years from May 2008 to January 2011 [25]. All patients were pretreated with chemotherapy and their Karnofsky PS was at least 80. The Charlson Comorbidity Index (CCI) calculator was used to record the comorbidities and it was 2–3 in nearly 70% of the subjects. Cetuximab 500 mg/m^2^ and irinotecan 180 mg/m^2^ were administered as a biweekly regimen until progression or unacceptable toxicity. Although grade 3 or 4 toxicities (skin rash 20%, diarrhea 18% and neutropenia 28%) were recorded, the authors confirmed that the combination of the two drugs was safe and effective in older patients with pretreated mCRC.

## 4. Anti-EGFR Inhibitors and Vunerable Patients

Until a few years ago, the SIOG recommendations for these subjects, who represent the preponderance of older patients, specified that chemotherapy could be administered with less intensive regimens through dose reduction and without excluding the use of oxaliplatin and irinotecan or resorting to the combination of bevacizumab and capecitabine [6]. The possibility of employing anti-EGFR in wt patients was not supported due to the substantial lack of data. Recent studies have reversed this position (Table 2).

### 4.1. PANDA

This phase II randomized prospective Italian study enrolled older patients with RAS and rapidly accelerated fibrosarcoma B (BRAF) wt mCRC. Each subject received either FOLFOX plus panitumumab for up to 12 cycles followed by panitumumab maintenance until disease progression (Arm A) or 5-FU/LV plus panitumumab for up to 12 cycles followed by panitumumab maintenance until disease progression (arm B) as first-line therapy [34]. The 5-FU bolus was not administered to contain toxicity in both arms. Among the inclusion criteria it was established a priori that patients aged 70 to 75 years had an ECOG PS of 1 or 2, while an ECOG PS of 0 to 1 was required for patients over 75 years of age. Collaterally, all patients underwent geriatric assessment through the G8 and the CRASH score evaluation in order to ascertain their vulnerability. The primary endpoint was PFS and the null hypothesis for this parameter was set at <6 months in both arms. Assuming a median expected PFS time of ≥9.5 months with both experimental regimens, a sample of 90 patients had to be enrolled in each arm to provide 90% power to the study, with a type I error rate of 5% for rejection of the null hypothesis.

From July 2016 to April 2019 a total of 185 patients were randomized (92 arm A and 93 arm B) and with a median follow up of 20.5 months, 135 (arm A/B: 64/71) progression disease (PD) events were collected. Although over 75% of patients experienced dose reductions or treatment delays in Arm A and more than 45% dose reductions and over 70% treatment delays in Arm B, the primary endpoint was met in both treatment arms (9.6 months in arm A and 9.1 months in arm B). Among the secondary endpoints, the RR was 65% and 57%, while the DCR was 88% and 86%, respectively, for arm A and B. The most relevant grade 3/4 toxicities were (arm A/B): neutropenia 9.8%/1.1%, diarrhea 16.3%/1.1%, stomatitis 9.8%/4.4%, neurotoxicity 3.3%/0%, fatigue 6.5%/4.4%, skin rash 25%/24.2%, and hypomagnesaemia 3.3%/7.7%. On this basis, the authors stated that 5-FU/LV plus panitumumab for up to 12 cycles followed by panitumumab maintenance until PD could be the treatment of choice for these patients. However, the final data of the study are yet to be released and it will be important to know the OS and the usefulness of translational analyses on tissue and blood samples of patients, the collection of which is still ongoing.

### 4.2. REVOLT

This retrospective study was based primarily on the consideration that planning a reduction in the doses of chemotherapy of two-drug regimens from the beginning of treatment results in greater tolerability without compromising the treatment effectiveness [40]. This strategy could be a winner when vulnerable old, RAS, and BRAF wt patients need to be treated with anti-EGFR that are known to increase chemotherapy toxicity. One hundred and eighteen patients were collected from 14 selected Italian centers in this observational trial [35]. The median age was 75 (range, 70–85). Although the most important limitation of the REVOLT study is to have identified the vulnerability of the patients based exclusively on a combined analysis of PS, weight, and type of comorbidity, physician’s clinical opinion, and use of the G8 (two thirds of cases had a score ≤11) without resorting to the better defined CGA, however, its retrospective and pragmatic nature should be emphasized since the patients were instead selected through the simple examination of medical records according to normal daily clinical practice. The primary endpoint was safety, and secondary endpoints were RR, PFS, and OS. Grade 3/4 toxicities appeared moderate in percentage and represented by neutropenia, found in only 11.8% of patients and rash in 11% of cases. The RR was in line with that of similar studies, while stable disease (SD) was observed in 29.1% of patients, with a DCR of 86.4%. With a median follow-up of 18 months, the median PFS and OS were 10.0 and 18.0 months, respectively. The authors concluded that by employing an appropriate design, including reduced doses from the first administration of chemotherapy, vulnerable older patients better tolerate a drug doublet regimen when combined with anti-EGFR antibodies.

### 4.3. PANEL

PANEL is a phase II study enrolling just 27 older patients (age ≥ 70 years) with wt RAS mCRC and treated with panitumumab plus capecitabine until disease progression or unacceptable toxicity as first-line therapy [36]. The majority of patients had an ECOG 1 (66.7%) and their median age was 78 years. The primary objective of the trial was RR, while the secondary objectives were DCR, duration of response (DoR), time to response (TTR), TTP and time to treatment failure (TTF), PFS, OS, and safety. The RR was 44.4% and, with a median follow-up of 17.7 months, the authors reported DCR in 19 (70.4%) patients, the median TTR was 2.2 months and the median DoR was 8.7 months. Median PFS from the start of treatment to data cut-off was 7.5 months and median OS was 23.7 months. Although capecitabine doses had been reduced (850 mg/m^2^ twice a day [bid] on days 1–14, every 3 weeks), the median duration of treatment was only around 20 weeks, the average number of panitumumab infusions was seven and that of capecitabine cycles was six. Doses of both drugs were reduced in one third of patients due to consistent toxicity in seventeen (63%) of the patients. The most frequent were skin toxicity (18.5%) and diarrhea (14.8%). Nine (33.3%) patients had serious AEs. Four (14.8%) of the patients died because of toxicity during the study. Nevertheless, the authors argued that, taking into account efficacy and safety parameters, capecitabine plus panitumumab may be considered to be a viable therapeutic option in older patients. These conclusions were rightly criticized as a treatment resulting in 63% of grade 3/4 toxicities and an almost 15% treatment-related mortality cannot be considered well-tolerated and interpreted as safe [41]. In addition, several studies have pointed out that the combination of an anti-EGFR and capecitabine cannot be selected as treatment of choice among all RAS and BRAF wt patients due to its well-known toxicity [32,33,39].

### 4.4. SAKK 41/10 Study

This multicenter, prospective randomized phase II trial should have investigated the benefit of cetuximab, either alone or in combination with capecitabine in vulnerable older patients with wt RAS and BRAF mCRC [39]. The inclusion criteria established that patients should be >75 years of age or ≥70 years with at least one additional adverse factor such as functional dependence or significant comorbidity. The primary endpoint was PFS at Week 12. The study was terminated prematurely due to slow accrual, despite the planned simple size of 78 patients. As mentioned above for studies analyzing the combination of anti-EGFR and capecitabine, a higher incidence of toxicities and treatment termination was observed. Skin rashes were common. Three patients reported ocular toxicity, in two of them of grade 3. Diarrhea, fatigue, infections, stomatitis, palmar-plantar erythrodysesthesia, and paronychia were more frequent in the combination arm. Two patients in this arm died after severe infection. At Week 12, 6 out of 11 patients (55%) were progression-free in the cetuximab monotherapy arm and 9 out of 13 patients (69%) in the combinaion arm. Due to the small sample size, the authors could only state that cetuximab monotherapy appeared tolerable and was preferred over the combination with capecitabine.

### 4.5. Jehn et al. Study

This German non-interventional study evaluated how tolerable and active ceuximab could be, mainly in combination with irinotecan (82%), in pretreated patients [37]. Among the more than 600 patients recruited from 87 centers, 305 patients were aged ≥65 years, with a median age of 71 years. PS was 1 or 2 in 78% of cases, with no difference between younger and older patients. The latter had major comorbidities and cardiovascular ones were the most frequently observed (31%). Patients received a median of 15 cetuximab infusions and were treated for an average of 4 months. Almost all patients (92%) had discontinued treatment before the end of the 12-month observation period, some times (23.6%) of their own will. Nonetheless, a considerable number of clinical responses were obtained with no differences between age groups. The RR was higher in patients treated without (50%) or with no more than one (45%) prior chemotherapy. Furthermore, PFS was not significantly different between younger and older patients. The toxicities appeared contained. Skin rash was more frequent (64%), but was rarely severe, and there was no significant difference based on age. Only 18.9% of patients experienced grade 3/4 toxicity. Older patients complained of a significantly longer duration of toxicities (9 days vs. 5 days for younger patients, *p* = 0.0004). A total of 6.3% of events led to permanent damage and 6.8% led to death, with no significant differences between the two age groups (*p* = 0.054). The most important limitation of this study was the lack of evaluation of the KRAS status, but it was not yet known how important this knowledge was when the trail started. However, the results are remarkable because they are obtained in a large number of patients, among other things pretreated.

### 4.6. OGSG 1602 Study

This Japanese phase II study enrolled 36 wt RAS older patients (median age, 81 years) defined as frail and not eligible for intensive doublet chemotherapy [38]. Thirty-four of the 36 subjects were evaluated for first-line efficacy with panitumumab, administered at a dose of 6 mg/kg intravenously every 2 weeks until the appearance of PD, unacceptable toxicities, patient withdrawal, physician decision, or surgical resection with curative intent. The primary endpoint was set as the DCR and disease radiological evaluations were performed every 8 weeks. Secondary endpoints were OS, PFS, RR, and the incidence of severe toxicities. Most patients (91.6%) had a PS of 0 or 1. Twenty-eight patients (77.8%) had a primary tumor location on the left. The RR was 50.0%, including three patients (8.8%) with complete responses. A total of 26.5% had SD, resulting in a DCR of 76.5%. The RR (65.4%) was only documentable for patients with left-sided tumors. Panitumumab was administered on average 8 times (range 1–16). Dose reduction by one level was necessary in eleven patients (30.6%) and four patients (11.1%) had doses reduced by two levels. The main grade 3 or 4 toxicities were rash (16.7%), hypomagnesaemia (11.1%), fatigue (8.3%), paronychia (5.6%), and hyponatremia (5.6%). A subsequent analysis of the data showed flattering results in terms of PFS and OS (6.0 and 17.5 months, respectively) [42]. Although the patients were not selected through the CGA, they could belong to the category of vulnerable older people because they were defined as potentially susceptible to chemotherapy, although not intensive. The authors implemented their selection through PS and clinical criterion considering that 16 patients (44.4%) had hypertension, 6 patients (16.7%) had diabetes, 5 patients (13.9%) had stroke, and 3 patients (8.3%) had ischemic heart disease. Panitumumab monotherapy showed favorable efficacy and feasibility in this setting of patients, especially for left-sided tumors.

## 5. Anti-EGFR Inhibitors and Frail Patients

These older patients are those for whom, either due to extremely advanced age and over 85 years or due to severe comorbidities or extremely poor PS, chemotherapy could not be recommended and who should undergo best supportive care [6]. However, at least two studies indicated that “frailty” was not an absolute criterion for excluding these patients from any form of treatment and a therapy with the use of only anti-EGFR was possible for palliative purposes.

### 5.1. Sastre et al. Study

This Spanish phase II study used panitumumab as the single agent and in first-line treatment for frail older patients with wt KRAS mCRC and poor prognostic factors [43]. Frailty was defined on the basis of the presence of ≥1 of the following characteristics: dependence for ≥1 basic activity of daily living (ADL); ≥3 comorbid conditions plus dependence for ≥1 instrumental ADL; the presence of ≥1 geriatric syndrome (age ≥85 years, fecal, or urinary incontinence in the absence of stress, frequent falls, spontaneous bone fractures, or neglect). The primary endpoint of the study was the 6-month PFS rate. A total of 33 patients were enrolled, their median age was 81 years and 19 (57.6%) of them had a PS of two. Panitumumab was administered on average for 8 cycles, while therapy was discontinued in 66.7% of cases for PD. The objective RR was 9.1%, while 18 patients (54.6%) achieved a SD with a DCR of 63.6%. The authors’ estimated 6-month PFS rate (30%) was achieved (36.5%) and the median OS was 7.1 months. In a subsequent analysis, after all RAS evaluation performed on 15 patients, the RR was 13.3%, the median PFS rate at 6 months was 53.3%, and the OS was 12.3 months. The most common adverse events were the expected ones and the most common significant grade 3 toxicity was rash (15.2%).

### 5.2. Pietrantonio et al. Study

A similar Italian phase II study evaluated panitumumab as the single agent in 40 frail older patients (aged ≥75 years) with wt RAS and BRAF mCRC [44]. The inclusion criteria allowed the enrollment of patients with absolute contraindication to any chemotherapy for first-line treatment or after failure of a fluoropyrimidine-based therapy for second-line treatment. The state of frailty was defined according to the Hurria criteria [45]. The primary endpoint was RR. The median age was 81 years (range 76–90) and PS was predominantly 1 (80%). Panitumumab was administered as first-line treatment in 10 (25%) patients and as second-line treatment in the remaining 30 (75%) patients. Among the 40 patients, 13 (32.5%) patients achieved a partial response, while 16 (40%) patinets had a SD. Therefore, the DCR was 72.5%. The primary endpoint of the study was met. Median PFS and OS were 6.4 months and 14.3 months, respectively. Although doses were reduced due to adverse events in nine (23%) patients, permanent discontinuation of treatment was never necessary. The most frequent grade 3 toxicities were skin rashes with an incidence of 20%, followed by fatigue and eye toxicity (2.5% each). The study confirms that anti-EGFR therapies are active in any line of treatment and above all that even frail older patients with wt RAS and BRAF mCRC could benefit from some form of active therapy without resorting to contraindicated chemotherapy, prolonging survival and improving the QoL of these patients.

## 6. Conclusions

This review was written with the aim of updating the results of studies with cetuximab or panitumumab with or without chemotherapy in older wt patients with mCRC. Although we have long had sufficiently consolidated data with the use of anti-EGFRs in the treatment of younger patients, the toxicities of these monoclonal antibodies make the potential benefits for older patients more confusing, also due to the lack of solid data and underrepresentation of this population in clinical trials.

However, at least for patients defined as fit or frail, international guidelines draw a fairly clear path on how these older patients could be treated with intangible benefits, QoL improvement, and toxicity containment and management.

The same cannot be said for vulnerable individuals who represent the majority of older patients. Consensus groups of geriatric oncology experts have defined how this group of subjects, at increased risk of toxicity and mortality, can be easily identified through the use of CGA or screening tools. The most recent studies reported above make it possible to state that, when using an anti-EGFR, this subcategory of patients could be treated either with a change of strategy involving their association with a planned dose reduction of a doublet chemotherapy from the first cycle or resorting to the administration of fluorouracil alone, however, avoiding the use of capecitabine.

Nevertheless, any approach to older patients will have to be cautious by overweighting their life expectancy and seeking the necessary involvement of their family members.

## Figures and Tables

**Table 1 jcm-11-07108-t001:** Anti-EGFR inhibitors and fit patients.

Line of Treatment	References	No. of pts	Treatment	Key Findings	Toxicities
First line	Opus/Crystal [17]	179	FOLFOX or FOLFIRI plus cetuximab vs. FOLFIRI or FOLFOX	Add Cetuximab improves RR (*p* < 0.0001), PFS (*p* < 0.0001), and OS (*p* < 0.0062), PFS and OS increased in older and younger pts	Higher toxicity and grade 3/4 AEs in pts > 70 years
First line	Prime [14]	1183	FOLFOX plus panitumumab vs. FOLFOX	PFS 10 vs. 8.6 months (*p* = 0.01), RR (57% vs. 48%, *p* = 0.02), OS (23.8 vs. 19.4 months, *p* = 0.03). PFS, HR 0.66 in pts < 65 years and 0.88 in pts ≥65 years. OS, HR 0.75 in pts <65 years and 0.80 in pts ≥65 years	Increase in grade 3/4 AEs with panitumumab arm: skin toxicity (37%), diarrhea (18%), hypomagnesemia (7%)
First line	FIRE 3 [18]	591	FOLFIRI cetuximab or FOLFIRI Bevacizumab	OS 33.1 vs. 25 months (*p* = 0.0059)	Grade 3/4 AEs: 64% for FOLFIRI Cetuximab vs. 51% FOLFIRI Bevacizumab
First line	ObservER study [19]	228	Chemotherapy cetuximab	OS 23.6 months, PFS 8.3 months	Grade ≥ 3 skin toxicity 14%
First line	ARCAD database [20]	1191	Doublet chemotherapy plus cetuximab or doublet chemotherapy alone	Advantage in PFS (11.2 vs. 8.9 months, *p* < 0.001) and in OS (23.9 vs. 20.3 months, *p* = 0.008) in younger, not for older pts	Older pts (vs younger) had no difference in grade 3/4 AEs
Pretreated	Bouchahda et al. study [21]	56	Cetuximab alone or with Irinotecan	RR 21%, PFS 4.4 months, OS 16 months. Similar efficacy to that observed in younger pts	Grade 3/4 AEs: Skin rash 11%, diarrhea 20%
Second line	Peeters et al. study [22]	600	FOLFIRI panitumumab or FOLFIRI	PFS 5.9 vs. 3.9 months, RR 35% vs. 10%, OS 14.5 vs. 12.5 months	Grade 3/4 AEs: 77% vs. 50% for pts < 65 years and 70% vs. 54% for pts ≥ 65 years
First line	EREBUS [23,24]	389	Cetuximab plus chemotherapy	PFS 9.5 vs. 9.2 months, RR 46.5% vs. 56.7%. No difference between older and younger pts	AEs of any grade no statistically different
Pretreated	Abdelwahab et al. study [25]	49	Cetuximab plus Irinotecan	Combination safe and effective in older pts with pretreated mCRC	Grade 3/4 AEs: skin toxicity 20%, diarrhea 18%, neutropenia 28%

FOLFOX, oxaliplatin, folinic acid, and 5-fluorouracil; FOLFIRI, irinotecan, folinic acid, and 5-fluorouracil; RR, response rate; PFS, progression-free survival; OS, overall survival; HR, hazard ratio; AEs, adverse events; pts, patients; mCRC, metastatic colorectal cancer.

**Table 2 jcm-11-07108-t002:** Anti-EGFR inhibitors and vulnerable patients.

Line of Treatment	References	No. of pts	Treatment	Key Findings	Toxicities
First line	PANDA [34]	185	FOLFOX plus panitumumab vs. 5-FU/LV plus panitumumab	PFS 9.6 vs. 9.1 months, RR 65% vs. 57%	Grade 3/4 AEs: neutropenia 9.8% vs. 1.1%, diarrhea 16.3% vs. 1.1%, skin toxicity 25% vs. 24.1%, stomatitis 9.8% vs. 4.4%
First line	REVOLT [35]	118	FOLFOX or FOLFIRI plus panitumumab or cetuximab	PFS 10 months, OS 18 months, RR 57.3%	Grade 3/4 AEs: neutropenia 11.8%, skin toxicity 11%
First line	PANEL [36]	27	Panitumumab plus capecitabine	RR 44%, PFS 7.5 months, OS 23.7 months	Grade 3/4 AEs: skin toxicity 18.5%, diarrhea 14.8%
Pretreated	Jehn et al. study [37]	305	Cetuximab alone or cetuximab plus Irinotecan	PFS not significantly different between younger and older pts	Grade 3/4 AEs: skin toxicity 18%, longer toxicity for older pts
First line	OGSG 1602 [38]	36	Panitumumab	PFS 6.0 months, OS 17.5 months, RR 50%	Grade 3/4 AEs: skin toxicity 16.7%, hypomagnesemia 11.1%, fatigue 8.3%, paronychia 5.6%
First line	SAKK 41/10 [39]	78	Cetuximab alone or cetuximab plus capecitabine	PFS at week 12: 55% vs. 69%	Diarrhea, fatigue, infections, stomatitis, HFS, and paronychia more frequent in combination arm

FOLFOX, oxaliplatin, folinic acid and 5-fluorouracil; FOLFIRI, irinotecan, folinic acid and 5-fluorouracil; RR, response rate; PFS, progression-free survival; OS, overall survival; AEs, adverse events; pts, patients; HFS, hand-foot syndrome.

## Data Availability

The data used to support the findings of this study are available from the corresponding author upon request.

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
