# Peer review of "An Update on the Role of Anti-EGFR in the Treatment of Older Patients with Metastatic Colorectal Cancer"

_jcm, 2022, doi:10.3390/jcm11237108_

Round 1
Reviewer 1 Report
Overall, I applaud the authors for this work. Entitled: An update on the role of anti-EGFR in the treatment of older patients with metastatic colorectal cancer
Comments to authors:
· Please make sure that the structure for citing published literature in the text, as well as the style of references in the References section, are consistent with the journal's style (see Instructions to Authors).
· English language needs revision for style and syntax.
· Abstract must be rewritten.
Author Response
Overall, I applaud the authors for this work. Entitled: An update on the role of anti-EGFR in the treatment of older patients with metastatic colorectal cancer
Comments to authors:
- Please make sure that the structure for citing published literature in the text, as well as the style of references in the References section, are consistent with the journal's style (see Instructions to Authors).
- English language needs revision for style and syntax.
- Abstract must be rewritten.
Reply to Reviewer 1:
- The structure and bibliographic references conform to the journal’s style.
- The text, as in the previous version, has been read and revised by native speakers.
- The abstract has been rewritten and streamlined, as have the conclusions. Each paragraph reporting the single studies has been revised trying to avoid as much as possible the duplication of sentences reported in other manuscripts.
Reviewer 2 Report
In this Review, the authors surmised the clinical studies on anti-EGFR treatment in metastatic colorectal cancer. They clarified how cetuximab and panitumumab should be used in older patients depending on their status, considering their vulnerability and the potential toxicity of these drugs. It has novelty and significance to update the usage of EGFR antibodies in elderly patients with metastatic colorectal cancer, however, unfortunately, this paper was found with a high duplicate rate with the publication by Rosati et.al, 2016.
This unfortunately cannot be published unless the authors rewrite the manuscript. There’re a couple of other suggestions:
1. Please extend the conclusion and discussion with more creative thinking and summarization, instead of just listing all the clinical studies.
2. It would also be good to underline the novelty and importance of the review in the abstract and in the conclusion.
Author Response
In this Review, the authors surmised the clinical studies on anti-EGFR treatment in metastatic colorectal cancer. They clarified how cetuximab and panitumumab should be used in older patients depending on their status, considering their vulnerability and the potential toxicity of these drugs. It has novelty and significance to update the usage of EGFR antibodies in elderly patients with metastatic colorectal cancer, however, unfortunately, this paper was found with a high duplicate rate with the publication by Rosati et.al, 2016.
This unfortunately cannot be published unless the authors rewrite the manuscript. There’re a couple of other suggestions:
- Please extend the conclusion and discussion with more creative thinking and summarization, instead of just listing all the clinical studies.
- It would also be good to underline the novelty and importance of the review in the abstract and in the conclusion.
Reply to Reviewer 2:
- We disagree on the consideration that many data follow what was already reported in a our previous review (JGO, 2016). Our manuscript arises from the need to update data on a very controversial and unclear topic: the anti-EGFRs treatment of older patients. In light of more recent studies that have clarified the role of cetuximab and panitumumab especially in vulnerable patients, it seemed necessary to offer a new overview of the literature data. Obviously, one cannot observe the present without forgetting the past.
- The abstract has been rewritten and streamlined, as have the conclusions. Each paragraph reporting the single studies has been revised trying to avoid as much as possible the duplication of sentences reported in other manuscripts.
- The concusion gives a more precise address that can be given to the reader on how an older patient should be treated with the anti-EGFRs. It seems to us a clear and important take-home message.
Round 2
Reviewer 2 Report
NA